# Environmental Features Influence Walking Speed: The Effect of Urban Greenery

**Marek Franěk \***  **and Lukáš Režný** 

Faculty of Informatics and Management, University of Hradec Králové, Rokitanského 62, 500 03 Hradec Králové, Czech Republic; lukas.rezny@uhk.cz
**\*** Correspondence: marek.franek@uhk.cz; Tel.: +420-49-333-2374

**Abstract:** The study investigated the rarely addressed topic of how visual environmental features can influence walking speed. Young adult participants were asked to walk on a route that leads through areas composed of urban parks and areas with a built environment with a large amount of greenery. Their walking speed was measured in selected sections. The participants walked with a small video camera, and their walk was recorded. The temporal information was derived from the video recordings. Subsequently, the participants evaluated the environmental features of the route by specific spatio-cognitive dimensions of environmental preference. The results show that walking speed in specific sections of the walking route systematically differed and reflected the environmental features. The walking speed was lower in sections with high natural characteristics and a high environmental preference. Noise here was perceived as less annoying than in sections with lower natural characteristics. The results are explained in terms of approach avoidance behavior. The findings are in accordance with environmental preference research that documents various benefits of walking in the natural environment.

**Keywords:** walking speed; greenery; urban environment; stress; relaxation

## 1. Introduction

Recently, physical inactivity has been a major public health problem [1]. Regular walking is a way of increasing daily physical activities. Walking is healthy, everyday moderate-intensity physical activity and has positive consequences for physical health (e.g., [2,3]). Walking is a basic physical activity that is easy to implement and is affordable and inexpensive. Regular walks are safe, and unlike other sports, there is practically no risk of injuries. Moreover, urban walking is a sustainable and environmentally friendly behavior. In recent decades, there has been increasing interest in research on walkability and walkable environments (for a review, see [4,5]). However, a fast pace of life is occurring in large cities, which includes a fast walking speed (e.g., [6,7]). Recent data suggest that the walking speed in large cities may gradually increase [8] as a response to the increasing level of urban stressors, such as noise [9]. Furthermore, it has been shown that a fast pace of life may be associated with coronary disease [10]. Therefore, it is important to focus attention on urban environmental stressors and explore environmental features that may reduce their impacts. Thus, the study is aimed to investigate how urban greenery may influence walking speed.

Walkability research is focused on the diverse environmental features of a walkable environment. However, a rarely addressed topic is how visual environmental features can influence walking speed. Given that a fast pace of life in large cities may be a potential health problem, it would be useful to explore environmental features that may affect walking speed. This study aimed to link walkability research with investigations of environmental preferences and psychological reactions to the natural environment and to explore walking speed as a response to the visual features of a walking environment.

### 1.1. Walkability and Benefits of Walking

Walkability is usually defined as the extent to which urban areas are easily walkable and friendly to people who walk or a measure of how convenient the built environment is to walk to predict levels of walking in a specific urban area [5]. Walkability typically involves various physical characteristics of an area, such as accessibility, pleasurableness, greenspace, diversity of land use, dwelling density, traffic safety, the number of intersections, and street connectivity [11,12]. Walkability is related to indicators of higher physical activity [13,14], lower obesity [15–17], and physical health [18–21].

Although there is general agreement on the benefits of walking in general, research within the field of environmental psychology provides further evidence of the positive effects of walking in a natural setting or urban nature compared to built settings without natural elements. Research has documented psychophysiological stress recovery and affective benefits after a walk in a natural setting compared to walking in an urban setting [22–25], improvement in cognitive function immediately after a natural walk [22,23,25–27], and even more positive body image after a natural walk [28]. Thus, the overall health benefits of urban walking can be increased with positive psychological effects if individuals walk in a natural setting or urban environment that contains some natural elements.

### 1.2. Walking Speed and Environmental Features

Previous research has shown that urban walking speed may be influenced by cultural, economic, and social factors (e.g., [29–31]) or atmospheric conditions [32–34]. Another line of research investigated capacity and pedestrian density [35–37] and modeled pedestrian traffic (e.g., [38–40]). Bosina and Weidmann's [41] review of existing literature investigating pedestrian walking speed summarized variables that influence walking speed: physical pedestrian properties (e.g., age, gender, body height), cultural differences (e.g., size of the settlement areas, cultural environment), emotional influences (e.g., mood, time pressure, trip purpose), and environmental influences (e.g., time of day, temperature pedestrian density, attractiveness of the environment). Although these authors considered the attractiveness of the environment within environmental influences, this factor has not been adequately explored. Thus, our previous research was devoted to features of the physical environment associated with attractiveness that may influence walking speed. Specifically, we explored how the level of naturalness in urban walking routes may influence this behavior.

It was shown [42] that the walking speed within a route changed in relation to specific environmental features of the walking route in an urban environment. In general, walking speed was faster in environments without any greenery and with traffic, while walking speed was slower in environments with greenery and lower levels of traffic. The slowest walking speed was found in alleys with mature old trees. The areas where participants walked slowly were also those they liked most. Importantly, these findings were observed in diverse vegetation periods, in spring and fall. A similar tendency was observed in a further study conducted on a different walking route [43]. Walking speed was slowest in sections with a higher level of greenery. Curiously, walking speed may accelerate immediately near ugly but not dangerous environments (e.g., dirty and damaged facades of houses, graffiti on a wall). In our subsequent study [44], participants listened to music at various tempi while walking on an urban route. It was found that fast music increased walking speed, but for both fast and slow music, participants walked more slowly in sections with a high amount of greenery than in other sections. In the next study [45] that was conducted with a partly similar route as in the previous experiment [44], the participants listened to traffic noise through headphones while walking. Traffic noise increased their walking speed compared to the control condition. However, participants' walking speed was also slower in the more natural sections than in the less natural sections while listening to the traffic noise and under the control condition. Thus, it seems that a level of perceived naturalness along a route is a crucial factor that decreases walking speed. It affects walking speed more than unpleasant acoustical stimuli, such as traffic noise.

### 1.3. Approach-Avoidance Behavior

The described effects of specific environmental features on walking speed have been explained [42] in terms of the theory of approach-avoidance behavior, which was elaborated within the field of environmental psychology by Mehrabian and Russell [46]. However, the behavioral approach-avoidance distinction is a more general concept in psychology. Approach and avoidance behavior can be seen as the fundamental building blocks that allow individuals to react and adapt to changing environmental properties [47]. It is considered to be a basic process that includes arousal and valence [48] and operates across various levels, from basic reflexes to cortical processes and social behavior [49]. Mehrabian and Russell [46] used these two forms of distinctive behaviors in the analysis of the affective reaction to a physical environment. Approach behavior refers to behaviors when an individual is trying to establish contact with the environment and remain inside it. It consists of diverse types of behavior: physical movement towards the environment, attention to the features of the environment, environment exploration, positive attitude towards the environment, and an attempt to perform any kind of activity in the environment. In contrast, behavioral reactions corresponding to avoidance behavior consist of an intention to attempt to avoid contact with the environment and to move quickly away. Thus, the approach to an environment may result in the deceleration of walking speed, while avoidance may result in the acceleration of walking speed.

This theory by Mehrabian and Russell [46] has been extensively tested in consumer research (for a review, see [50,51]), specifically in studies that investigated the effects of various dimensions of retail environments (e.g., temperature, noise, music, layout, equipment, signs, symbols) on customer reactions. Our previous study [42] explored whether the approach-avoidance model can be used to explain changes in walking speed in sections with different environmental features. The results showed that acceleration in walking speed in sections of a route with a relatively low amount of greenery and a high level of traffic and noise was linked with avoidance behavior (measured by dimensions of pleasure, arousal, and dominance from the Mehrabian-Russell questionnaire), while deceleration of walking speed occurred in sections with a high amount of greenery and a low level of traffic and noise.

### 1.4. Preferred Environment

A large body of research has shown that natural environments containing vegetation are preferred over urban and human-made environments, and in the urban setting, environments that contain some amount of vegetation are preferred (e.g., [52–58]). Within the field of environmental psychology, there have been attempts to explain environmental preference using cognitive analyses of visual features of an environment and to define the cognitive dimensions that influence it. Kaplan [59], in his pioneering work, proposed that environmental preference is based mainly on four cognitive dimensions: *coherence*, *complexity*, *legibility*, and *mystery*. A higher level of perceived coherence, complexity, legibility, and mystery could result in a higher environmental preference. Herzog [56,60] subsequently elaborated a cognitive analysis of the preferred environment. In addition to the variables of coherence, readability, complexity, and mystery from Kaplan's model, he established further cognitive dimensions. For instance, *spaciousness* is related to the spatial configuration of the environment. This variable describes the perception of sufficient space in the environment for the possibility of free and unrestricted movement within the environment. Another dimension is *nature*, the number of natural elements. The more natural elements that are present in the environment, the more preferred it is. These cognitive dimensions have been widely used in subsequent environmental preference investigations (e.g., [60–64]). Environmental psychology research has also established the concept of *restoration* of an environment, where a crucial role is played by cognitive visual dimensions (e.g., [60,65]).

*1.5. The Present Study*

The objective of the present study was to investigate the effect of visual environmental features on pedestrian walking speed. The previous studies [42–44] documented the faster walking speed in urban environments without any greenery and with traffic in contrast to slower walking speed in urban environments with greenery and lower traffic levels. However, these studies contrasted urban natural environments with environments without greenery, but we do not know whether people react similarly to different amounts of greenery in urban environments. To better understand these mechanisms it is also necessary to know how people perceive the environment they walk through and whether their walking behavior reflects their perception of the environment, specifically, their environmental preference and liking the environment. The change of walking speed as a response to a visual environment with a specific function of greenery may be a specific and underexplored example of specific sensorimotor reactions. Better knowledge about these mechanisms can advance scientific knowledge in the field. Moreover, understanding this problem more deeply might help architects and urban designers to create a more sustainable urban design.

Thus, we selected a walking route that leads through areas composed of urban parks and areas with a built environment but a large amount of greenery. Study 1 analyzes walking speed in eleven selected sections of this route. In Study 2, the participants evaluate the environmental features of the walking route by specific spatio-cognitive dimensions of environmental preference. The effect of the purpose of the walk on walking speed is explored in Study 3. It is hypothesized that walking speed will decelerate in the sections of the route with a high environmental preference. Furthermore, it is expected that the tendency to slow down in the preferred environment and speed up in the environment with a low level of preference will be more pronounced when people can actively observe their surroundings than when people go through the environment without paying attention to it.

## 2. Study 1

The goal of this study was to explore the effect of environmental features on walking speed. The participants were asked to walk around urban routes with different amounts of greenery. Their walking speed was measured in eleven selected sections of the walking route.

*2.1. Materials and Methods*

2.1.1. Participants

A total of 105 undergraduates attending psychology courses participated in the study. The students were young adults aged 19 to 28 years ($M$ = 20.96 years, $SD$ = 1.23), and the sample was composed of 52 men and 53 women.

2.1.2. Walking Route

The study was conducted in the central area of Hradec Králové. This city has approximately 100,000 inhabitants. The walking route was a circuit with a length of 1.75 km. The route was located on a flat surface with asphalt pavement. The route was chosen because it did not to run alongside other visually distinctive visual features, such as buildings with distinctive architecture, distinctive urban mobiliary, shops, advertisements, etc. Walking speed was measured on eleven selected sections (Figure 1). The sections were chosen to provide a direct path and avoid crossing crossroads or other obstacles.

Section 1 (60 m) and section 2 (60 m)—The walking path leading through a small park. On the left is a river, and on the right of the road behind are mature trees.

Section 3 (100 m) and section 4 (100 m)—The walking path approached the road from the right side. On the left side there are mature trees, behind them is a hillside overgrown with grass, and behind that are a meadow and a river.

Section 5 (60 m), section 6 (80 m) and section 7 (90 m)—The walking path leads through a dense oak alley with mature trees. On the left there is a meadow leading to the river, and on the right, there are low trees.

Section 8 (80 m)—A city street. There are villas with gardens on the left, a strip of greenery and mature trees on the right, and a road behind them.

Section 9 (60 m)—A city street. There is a small park on the left and a school building behind it, a road on the right and houses behind it.

Section 10 (80 m)—A city street. On the left side behind the school building, there is a view of a park with tall old trees, a road on the right, and houses behind it.

Section 11 (100 m)—A city street. On the left side, there is a moat near the sidewalk, and behind it, there is a long low building (former barracks); on the right side, there is a strip of greenery with low landscaped trees, with a road and houses behind them. There is a busy intersection nearby.

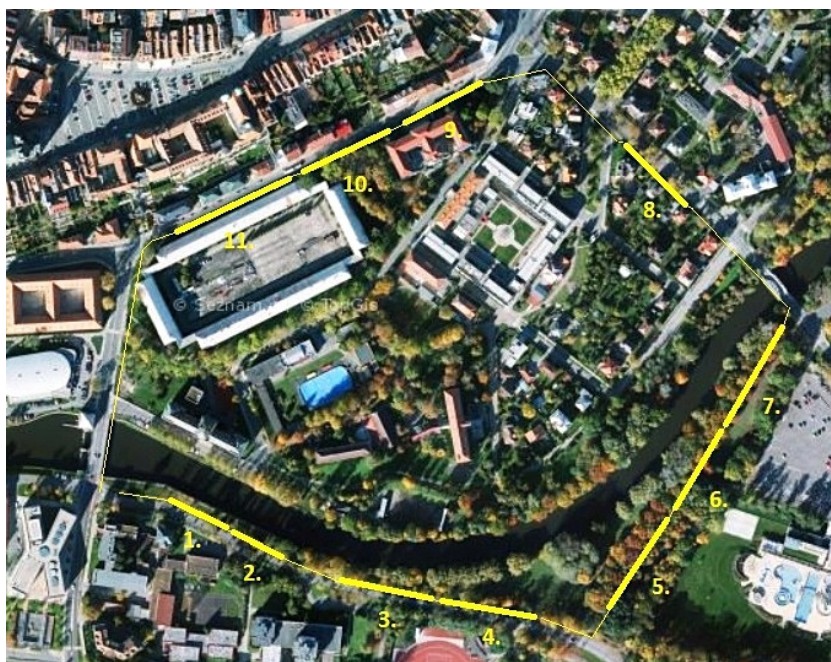

**Figure 1.** Walking route. Sections 1–11, where walking speed was measured, are indicated.

### 2.1.3. Measurement of Walking Speed

The participants walked with a small video camera (Sony Bloggie MHS-PM5K, Sony Corporation, Tokyo, Japan) attached to a belt around their waist (size $19 \times 108 \times 55$ mm$^3$, weight 110 g). Through a fisheye lens, the environment, the participant's feet, and the participant's arms were captured. The beginning and end of each section of the route were indicated by a line drawn on the sidewalk. An evaluator marked two frames of the video recording to create the beginning and end of the annotation for each section in the software Elan (see https://tla.mpi.nl/tools/tla-tools/elan/ (accessed on 15 April 2021). Every annotation represented the entire section of the track so that the extent of time subjects spent there could be determined. This enabled us to calculate the average speed reached by the participants in specific sections. The data are in Supplementary Materials, Table S1.

### 2.1.4. Procedure

Before the experiment, the participants were asked to sign informed consent forms. Then, they were informed that they would take part in a study in which their task was to walk around a route. The route was shown and explained to them using a map. They were instructed to walk through the route at their normal walking speed. Furthermore,

they were asked to not stop walking and not to call or speak with other pedestrians. They were not informed about the goal of the study. They then received a belt with a video camera and walked approximately 400 m to the initial point of the route. Here, the research assistant switched their camera on and showed them the direction of their walk. The participants walked individually around the route. The walk lasted approximately 20 min. The walking route was marked by orange arrows painted on the sidewalk. To control for a possible effect of the order of individual sections of the route, the participants walked on the route either from section 1 to section 11 (direction 1–11) or from section 11 to section 1 (direction 11–1). The participants were randomly assigned to a specific condition. In direction 1–11, 53 participants walked (26 males, 27 females), in direction 11–1, 52 participants walked (26 males, 26 females).

The study was conducted in April 2013 on three workdays: 23 April (only afternoon), 24 April and 25 April. On the afternoon of 23 April, it was cloudy, with mild wind and a temperature of approximately 20 degrees Celsius. On the morning of 24 April, it was sunny, and the temperature was approximately 20 degrees. The afternoon of 24 April was sunny and had a temperature of approximately 23 degrees. On the morning of 25 April, it was cloudy, and the temperature was approximately 22 degrees. On the afternoon of 25 April, it was cloudy, and the temperature was approximately 25 degrees. The grass was green, bushes were bright green, and trees along the route were sparsely covered with leaves.

### 2.2. Results and Discussion

Table 1 and Figure 2 show the average walking speeds in the individual sections. The average walking speed in directions 1–11 was 1.62 m/s (SD = 0.14) and was faster than the average walking speed in directions 11–1, which was 1.59 m/s (SD = 0.15). However, the difference was not significant ($t = 1.29$, $p = 0.20$). To test the assumption that the environmental characteristics of specific sections and the direction of the walk affect the walking speed, a mixed analysis of variance (ANOVA) was conducted to test the effects of individual sections (11 sections) and the direction of the walk (direction 1–11—direction 11–1) on walking speed in individual sections of the route. The direction of the walk on the route was chosen as the between-subject factor, the section of the route was chosen as the within-subject (repeat-measures) factor, and the speed of walking was used as the dependent variable. The calculations were conducted by the software *Statistica 12* (TIBCO Software Inc., Palo Alto, CA, USA). ANOVA indicated a statistically significant within-subjects main effect of the section walked ($F_{10,102} = 19.40$, $p < 0.001$, $\eta2 = 0.16$). A post hoc Tukey test showed that the participants' walking speeds differed statistically significantly in the following pairs of sections:1–5, 1–6, 1–7, 2–4, 2–5, 2–6, 2–7, 3–5, 3–6, 3–7, 4–5, 4–6, 4–7, 4–9, 4–11, 5–8, 5–9, 5–10, 5–11, 6–8, 6–9, 6–10, 6–11, 7–8, 7–9, 7–10, and 7–11. The between-subjects main effect of the direction of the walk was not significant ($F_{1,102} = 1.65$, $p = 0.20$, $\eta2 = 0.02$). The interaction between the section walked and the direction of the walk was significant ($F_{10,102} = 6.61$, $p < 0.001$, $\eta2 = 0.06$).

**Table 1.** The average walking speeds (m/s) in specific sections of the route in Study 1.

| Section | Direction 1–11 | | Direction 11–1 | |
|---|---|---|---|---|
| | **M** | **SD** | **M** | **SD** |
| Section 1 | 1.65 | 0.12 | 1.58 | 0.15 |
| Section 2 | 1.65 | 0.14 | 1.59 | 0.14 |
| Section 3 | 1.64 | 0.14 | 1.58 | 0.15 |
| Section 4 | 1.62 | 0.14 | 1.59 | 0.15 |
| Section 5 | 1.60 | 0.15 | 1.57 | 0.15 |
| Section 6 | 1.59 | 0.15 | 1.57 | 0.15 |
| Section 7 | 1.59 | 0.15 | 1.57 | 0.15 |
| Section 8 | 1.62 | 0.15 | 1.60 | 0.15 |
| Section 9 | 1.63 | 0.15 | 1.60 | 0.14 |
| Section 10 | 1.62 | 0.15 | 1.60 | 0.15 |
| Section 11 | 1.62 | 0.15 | 1.62 | 0.15 |
| Average speed on the route | 1.63 | 0.15 | 1.59 | 0.15 |

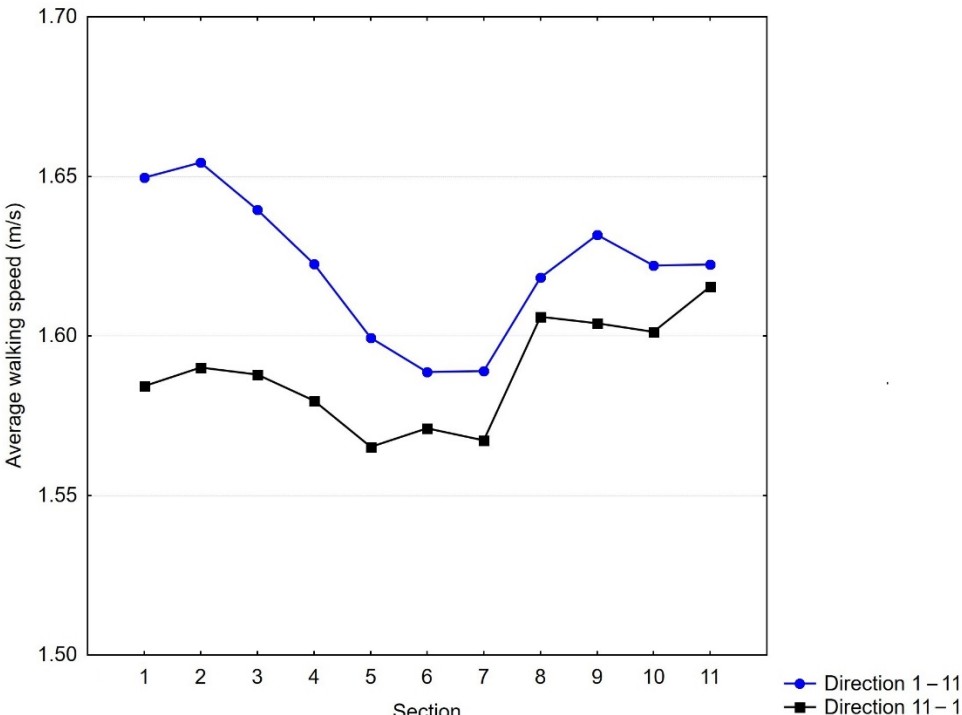

**Figure 2.** Average walking speeds (m/s) in specific sections of the route in Study 1.

The results showed that the walking speed in individual sections differed. The walking speeds in sections 5, 6, and 7 were significantly slower than the walking speed in other sections. sections 5, 6, and 7 were located in the dense oak alley, which was surrounded on one side by a meadow and on the other side by vegetation. Clearly, this part of the route had higher natural character than the other sections of the route. Moreover, there were observed differences in walking speed between section 10 and the previous section (section 9 in walking direction 1–11 and section 11 in walking direction 11–1), but these differences were not significant (see Figure 2). Section 10 was characterized by a sudden change in the environment, in contrast to the previous sections, which consisted of an open space with a park and mature old trees.

The interaction between the section walked and the direction of the walk revealed that in the direction 11–1, walking speed decreased at the final parts of the route; in sections 1–4, the walking speed was significantly lower than in the direction 1–11. Importantly, the walking speed in section 11 was almost equal in both walking directions, regardless of whether this section was the first or the last section of the route. We speculate that in this section, the sidewalk was narrow, and there was a deep moat between its edge and the adjoining building, causing a decrease in walking speed. Thus, the participants who started in section 11 walked slightly slower than those who started the walk in section 1 and continued to section 11. It seems that the initial speed within the walking route influenced the speed on the whole route.

Although the route was chosen because it did not to run alongside other visually distinctive architecture, one exception may be section 11 that ran all along the long low building, former barracks build in a 18th-century military style that is not a common architectonical style in this city. On the other hand, the building had a very simple facade with minimal architectural details that did not catch attention, we suppose. Moreover, the data did not show any systematic effect of this architecture on walking behavior.

## 3. Study 2

The goal of this study was to evaluate the environmental features of eleven sections of the walking route from Study 1. The environments were evaluated by spatio-cognitive

dimensions based on studies by Kaplan [59] and Herzog [56,60]. Moreover, perceived noise, traffic, and amount of natural elements were assessed. Finally, the participants' liking of the environment and intention to leave the environment (approach-avoidance behavior) were assessed.

### 3.1. Materials and Methods

#### 3.1.1. Participants

A total of 72 undergraduates attending psychology courses participated in the study. The students were young adults aged 19 to 23 years ($M$ = 20.81 years, $SD$ = 0.90), and the sample was composed of 35 men and 37 women. All of them took part in Study 1.

#### 3.1.2. Measures

On the eleven sections of the walking route employed in Study 1, the participants were required to rate the environment using eleven items on the questionnaire (see Table 2).

**Table 2.** The questionnaire used in Study 2.

| | Item | Dimension |
|---|---|---|
| | | Spatio-cognitive dimensions |
| 1. | There is enough space to move around sufficiently and without restrictions | Space |
| 2. | The individual elements of the environment are in harmony with each other. | Coherence |
| 3. | It's easy to understand where I am now, what the place is, how to get out of here and where to go next. | Legibility |
| 4. | The environment contains a large number of various elements. | Complexity |
| 5. | If I enter the environment and move on, it is possible that I will find many other interesting things there. | Mystery |
| 6. | The environment offers relaxation, calming, and an escape from everyday stress. | Relaxation |
| | | Environmental features |
| 7. | There is a lot of traffic. | Traffic |
| 8. | There is noise here. | Noise |
| 9. | There are a lot of natural elements here. | Naturalness |
| | | Approach-avoidance behavior |
| 10. | I prefer to leave this place | |
| | | Liking |
| 11. | I like the environment. | |

Items 1–6 described spatio-cognitive dimensions of the environment. These items were selected from the study by Herzog [60]. Item 1 was related to the spatio-cognitive dimension *space*, item 2 was related to the spatio-cognitive dimension *coherence*, and item 3 was related to the spatio-cognitive dimension *legibility*. Item 4 was related to the spatio-cognitive dimension *complexity*, item 5 was related to the spatio-cognitive dimension *mystery*, and item 6 was related to the dimension *restoration.* Items 7–9 described properties of the environment. Item 7 was related to perceived *traffic*, item 8 was related to perceived *noise*, and item 9 was related to the perceived amount of natural elements (*naturalness*). Item 10 described *intention to leave the environment* (avoidance behavior), *and* item 11 described the participants' *liking* of the environment. The level of agreement/disagreement with these items was expressed on a 5-point Likert-type scale with anchors 1: Absolutely disagree and 5: Absolutely agree. The data are in Supplementary Materials, Table S1.

#### 3.1.3. Procedure

The participants walked together in a group with the author of the study along the walking route that was employed in Study 1. In approximately the middle of each

section of the walking route, the participants stopped and recorded their evaluations on a questionnaire form. To prevent participants from influencing each other in their evaluation of an environment, they were asked not to talk to each other while completing the questionnaire. The first group of participants (N = 34, 20 females) moved along the route from section 1 to section 11, while the second group of participants (N = 38, 17 females) moved along the route from section 11 to section 1. The form of vegetation and weather was approximately the same as at the time of Study 1. The study was conducted during one working day, 2 May 2013.

### 3.2. Results and Discussion

The scores of the estimation of perceived spatio-cognitive dimensions of the specific section of the walking route are shown in Figure 3a. The scores of the estimation of perceived environmental properties of the specific section of the walking route are shown in Figure 3b, and the scores of the intention to leave the environment and liking of the environment are shown in Figure 3c.

To test the assumption that the environmental features of specific sections affect perceived spatio-cognitive dimensions, perceived environmental properties, intention to leave the environment (approach-avoidance behavior), and liking the environment (see Table 3), a series of one-way repeated measures ANOVAs were conducted. The preliminary analysis did not reveal differences between the estimation of participants moving from section 1 to section 11 and those moving from section 11 to 1. Therefore, the direction of the walk on the route was not included in the ANOVA model. The calculations were conducted by the software *Statistica 12* (TIBCO Software Inc., Palo Alto, CA, USA).

*Space.* The one-way repeated measures ANOVA showed that the environmental features of specific sections had a moderate effect on the level of perceived space. The significantly highest levels of perceived space were in sections 5, 6, and 7 (compared to sections 3, 4, 8, 9, 10, 11), and section 10 had significantly higher perceived space compared to surrounding sections 9 and 11.

*Coherence.* The one-way repeated measures ANOVA showed that the environmental features of specific sections had a moderate effect on the level of perceived coherence. Sections 5, 6, and 7 had the highest level of perceived coherence compared to the other sections. In contrast, the significantly lowest level of perceived coherence was found in section 11 compared to the other sections (except sections 10 and 1).

**Table 3.** Results of the ANOVA for all questionnaire items used in Study 2.

| Item | F | *p* | η2 | df |
|---|---|---|---|---|
| **Spatio-cognitive dimensions** | | | | |
| 1. Space | 39.27 | 0.001 | 0.34 | 1,10 |
| 2. Coherence | 46.72 | 0.001 | 0.40 | 1,10 |
| 3. Legibility | 9.86 | 0.001 | 0.12 | 1,10 |
| 4. Complexity | 10.82 | 0.001 | 0.13 | 1,10 |
| 5. Mystery | 7.31 | 0.001 | 0.09 | 1,10 |
| 6. Relaxation | 104.932 | 0.001 | 0.60 | 1,10 |
| **Environmental properties** | | | | |
| 7. Traffic | 264.43 | 0.001 | 0.79 | 1,10 |
| 8. Noise | 179.36 | 0.001 | 0.72 | 1,10 |
| 9. Nature | 100.88 | 0.001 | 0.60 | 1,10 |
| **Approach-avoidance behavior** | | | | |
| 10. Leave | 64.66 | 0.001 | 0.48 | 1,10 |
| **Environmental preference** | | | | |
| 11. Liking | 75.45 | 0.001 | 0.52 | 1,10 |

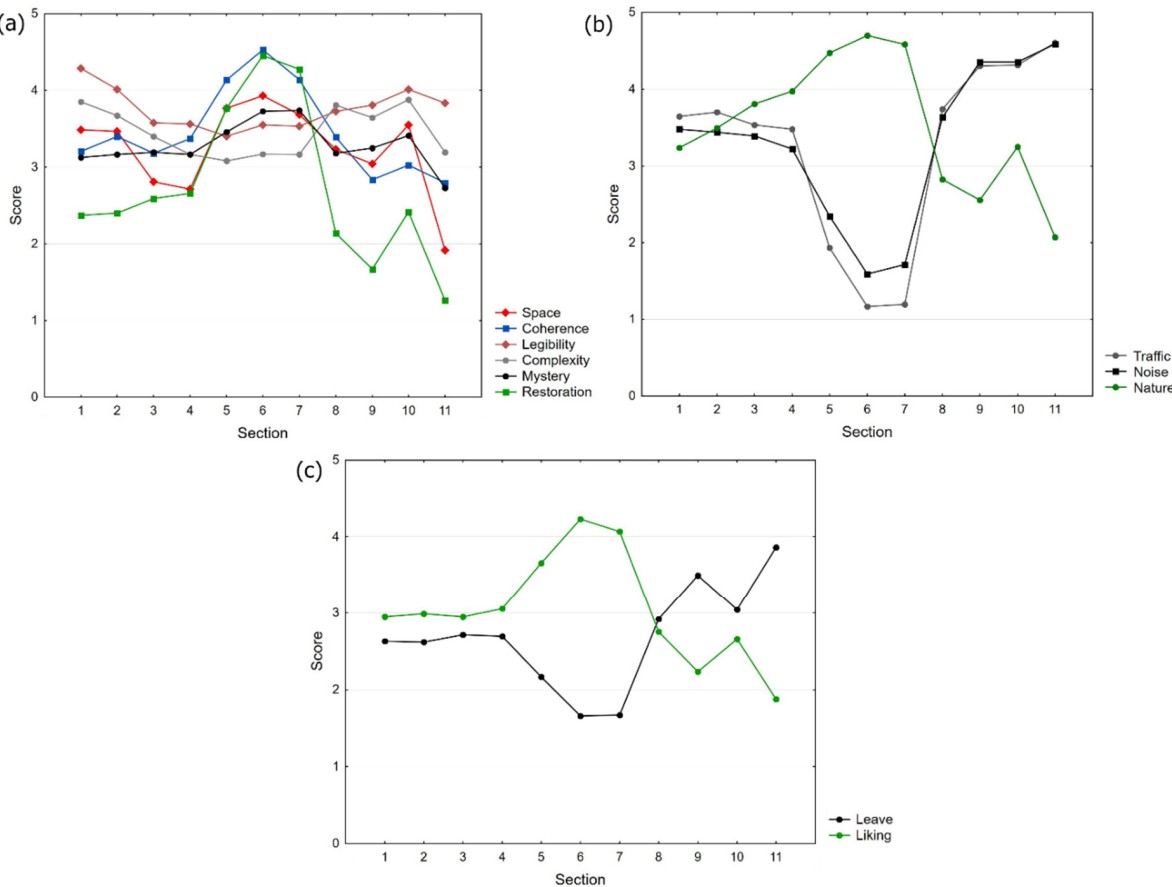

**Figure 3.** Average scores for spatio-cognitive dimensions (**a**), properties of the environment (**b**) and intention to leave the environment and liking (**c**).

*Legibility.* The one-way repeated measures ANOVA showed that the environmental features of specific sections had a small effect on the level of perceived legibility. A significantly higher level of perceived legibility was found for sections 1 and 2 compared to sections 5, 6, 7, 9, and 11 and for section 10 compared to sections 5, 6, and 7. A significantly lower level of legibility was found in section 11 than in the other sections (except section 10).

*Complexity.* The one-way repeated measures ANOVA showed that the environmental features of specific sections had a small effect on the level of perceived complexity. A higher level of perceived complexity was found in sections 1, 2, and 8–10. The significantly lowest levels of perceived complexity were found in section 5 and section 6.

*Mystery.* The one-way repeated measures ANOVA showed that the environmental features of specific sections had a small effect on the level of perceived mystery. A significantly higher level of perceived mystery was found in section 6 and section 7 (except combinations 6–9, 6–10, 7–9, 7–10), and the significantly lowest level of perceived mystery was found in section 11 (compared to sections 5, 6, 7, 9 and 10).

*Restoration.* The one-way repeated measures ANOVA showed that the environmental features of specific sections had a high effect on the level of perceived relaxation. Sections 5, 6, and 7 had significantly higher levels of relaxation, and section 10 had a higher level of perceived relaxation than sections 9 and 11. The significantly lowest level of relaxation was observed in section 11 compared to the other sections (except combination 9–11).

*Traffic.* The one-way repeated measures ANOVA showed that the environmental features of specific sections had a moderate effect on the perceived level of traffic. Sections 5, 6, and 7 had significantly lower levels of perceived traffic than the other sections; in contrast, sections 9, 10, and 11 had significantly higher levels of perceived traffic than the other sections.

*Noise.* The one-way repeated measures ANOVA showed that the environmental features of specific sections had a high effect on the level of perceived surrounding noise. The results were similar to the perceived level of traffic in specific sections, indicating that noise is mainly generated by traffic.

*Naturalness.* The one-way repeated measures ANOVA showed that the environmental features of specific sections had a high effect on perceived naturalness. There was a significantly increased level of naturalness from section 3 to section 7, and section 10 had a significantly higher level of perceived naturalness compared to the surrounding sections 9 and 11.

*Intention to leave the environment.* The one-way repeated measures ANOVA showed that the environmental features of specific sections had a high effect on the intention to leave the environment. A significantly lower level of intention to leave the environment occurred in sections 5, 6, and 7 compared to other sections, while the significantly highest level of intention to leave the environment occurred in sections 9 and 11 compared to other sections.

*Liking the environment.* The one-way repeated measures ANOVA showed that the environmental features of specific sections had a high effect on liking the environment. Sections 5, 6, and 7 had significantly higher levels of environmental preference than the other sections, and section 11 had significantly lower levels of environmental preference.

The results indicate that the environmental features of specific sections of the walking route had significant effects on the perceived qualities of the environment. They had a significant and mediate effect on perceived *coherence*, *space*, and *restoration*. The effect of specific sections was almost the same among these spatiotemporal dimensions. The effect on perceived legibility, complexity, and mystery was also significant but smaller. Thus, *coherence*, space, and *restoration* seem to be the spatio-cognitive dimensions that best described the differences between various environments of the walking route. The environmental features of specific sections also had significant mediating effects on the perceived level of traffic and noise and a high effect on perceived naturalness. These results are consistent with the scores of perceived spatio-cognitive dimensions and liking scores. Curiously, the scores of the spatio-cognitive dimensions *legibility* and *complexity* that are supposed to be combined with the preferred natural environment (Kaplan, 1975) showed an opposite trend to the scores of *coherence*, *space*, *mystery* and *restoration*.

The results consistently showed a specific character of sections 5, 6, and 7, which are composed of a long dense oak alley. These sections had the highest scores for perceived naturalness and liking. Accordingly, the spatio-cognitive dimensions of environmental preference *coherence*, *space*, *mystery*, and *restoration* had the highest scores in these sections. Moreover, perceived noise and traffic were the lowest here. Another section with a similar specific character is presented in section 10 with a view of the park with tall old trees.

Importantly, these findings reflect changes in walking speed on the route observed in Study 1. The slowest walking speed was in sections 5, 6, and 7, which had the highest level of liking and perceived naturalness, a low level of noise and traffic, and high values of *coherence*, *space*, *mystery*, and *restoration*. The same holds for section 10 compared with the surrounding sections 9 and 10. The results also confirm that changes in walking speed may be explained in terms of approach-avoidance behavior. Perceived avoidance behavior (the intention to leave the environment) was lowest in the sections that were liked most and where walking speed was lower than in the other sections.

## 4. Study 3

In Study 1, the purpose of walking was not specified. In particular, the purpose of walking may be related to the extent to which the participants observe the surrounding environment, which may influence their walking speed. There is evidence that business and commuter traffic have the highest walking speeds, while leisure trips have the lowest walking speeds [41]. Therefore, in the present study, participants were given different instructions: either to observe the environment carefully while walking or to ignore the

environment and look ahead. The objective was to explore the extent to which the difference between active monitoring of the surroundings or trying to walk the route without monitoring the surroundings can affect walking speed in individual sections of the route. The study was conducted on the same route as in Study 1, but for practical reasons, the route length was shortened; it contained only the sections where the largest difference in walking speed was observed in Study 1.

### 4.1. Materials and Methods

#### 4.1.1. Participants

A total of 79 undergraduates attending psychology courses participated in the study. The students were young adults aged 19 to 25 years ($M$ = 21.11 years, $SD$ = 1.05), and the sample was composed of 38 men and 41 women.

#### 4.1.2. Walking Route

The walking route was partly the same as in Study 1, but it was shorter, encompassing only sections 1–8.

#### 4.1.3. Procedure

The procedure and measurement of walking speed were the same as in Study 1. There were two conditions: "look around" and "do not look around". Under the condition "look around", the participants received the following instruction: "*While walking along the route, look carefully around you. Try to remember what was interesting on the route*". In contrast, under the condition "do not look around", the participants received the following instruction: "*While walking along the route, try to ignore your surroundings and just look at the road ahead*".

The participants were randomly assigned to a specific condition. Under the condition *look around*, 41 participants walked (20 males, 21 females); under the condition *do not look around*, 38 participants walked (18 males, 20 females).

The study was conducted in April 2016 on three workdays, 26 April (only afternoon), 27 April, and 28 April. On the afternoon of 23 April, it was cloudy, with a temperature of approximately 10 degrees Celsius. On the morning of 24 April, it was cloudy, and the temperature was approximately 6 degrees. On the afternoon of 24 April, it was cloudy with a temperature of approximately 7 degrees. On the morning of 25 April, it was cloudy, and the temperature was approximately 7 degrees. On the afternoon of 25 April, it was cloudy, with a temperature of approximately 10 degrees. The grass was green, bushes were bright green, and trees along the route were sparsely covered with leaves.

### 4.2. Results and Discussion

Table 4 and Figure 4 show the average walking speeds in individual sections. The average walking speed under the condition *look around* was 1.60 m/s (SD = 0.14); this was slower than the average walking speed under the condition *do not look around*, 1.65 m/s (SD = 0.13). The difference was significant ($t$ = 2.19, $p < 0.05$, Cohen's $d$ = 0.37). To test the assumption that the environmental characteristics of specific sections and the condition of the walk affect the walking speed, a mixed ANOVA was conducted to test the effects of specific sections (8 sections) and the condition (*look around-do not look around*) on walking speed in individual sections of the route. The condition of the walk on the route was chosen as the between-subject factor, the section of the route was chosen as the within-subject (repeat-measures) factor, and the speed of walking was used as the dependent variable. The calculations were conducted by the software *Statistica 12* (TIBCO Software Inc., Palo Alto, CA, USA). ANOVA indicated a statistically significant within-subjects main effect of section walked ($F_{7,73}$ = 53.21, $p < 0.001$, η2 = 0.42). A post hoc Tukey test showed that the participants' walking speeds differed statistically significantly in almost all pairs of sections except sections 1–2, 2–3, 3–4, 3–8, 4–8, 5–6, and 6–7. Furthermore, ANOVA indicated a statistically significant between-subjects main effect of the condition

($F_{1,73}$ = 4.79, $p \leq 0.05$, η2 = 0.06). The interaction between the section walked and the condition was also significant ($F_{7,73}$ = 4.51, $p < 0.001$, η2 = 0.06).

**Table 4.** The average walking speeds (m/s) in specific sections of the route in Study 3.

| Section | Look around | | Do Not Look around | |
|---|---|---|---|---|
| | **M** | **SD** | **M** | **SD** |
| Section 1 | 1.62 | 0.14 | 1.67 | 0.13 |
| Section 2 | 1.62 | 0.15 | 1.67 | 0.12 |
| Section 3 | 1.60 | 0.15 | 1.66 | 0.13 |
| Section 4 | 1.59 | 0.15 | 1.66 | 0.12 |
| Section 5 | 1.55 | 0.15 | 1.63 | 0.13 |
| Section 6 | 1.54 | 0.14 | 1.63 | 0.13 |
| Section 7 | 1.52 | 0.14 | 1.62 | 0.13 |
| Section 8 | 1.58 | 0.15 | 1.66 | 0.12 |
| Average speed on the route | 1.60 | 0.14 | 1.65 | 0.13 |

The results showed that the instruction significantly influenced overall walking speed on the route. As expected, under the condition *do not look around*, the participants walked faster than under the condition *look around*. Interestingly, the walking speed under the *look-around* condition was approximately the same as that in Study 1. This finding shows that the instruction *do not look around* led to a change in behavior and reduced observation of the environment, which results in faster walking speed. However, regardless of the instructions and walking speed, the data showed the same trend under both conditions, specifically the significantly slowest walking speed in sections 5, 6, and 7 formed by an oak alley and a significant increase in walking speed in section 8 formed by a city street with a smaller level of greenery. The temporal pattern in walking speed for the route is in accordance with the pattern found in Study 1. It shows a very robust effect of environmental features on walking speed. The robustness of the trend is also supported by the fact that Study 3 was conducted at a substantially colder temperature and slightly denser vegetation than Study 1. The limitation was that the research methodology did not enable control of actual head movements and the direction of the eyes, but the significant differences between average speeds under both conditions indicate that the instruction was largely respected.

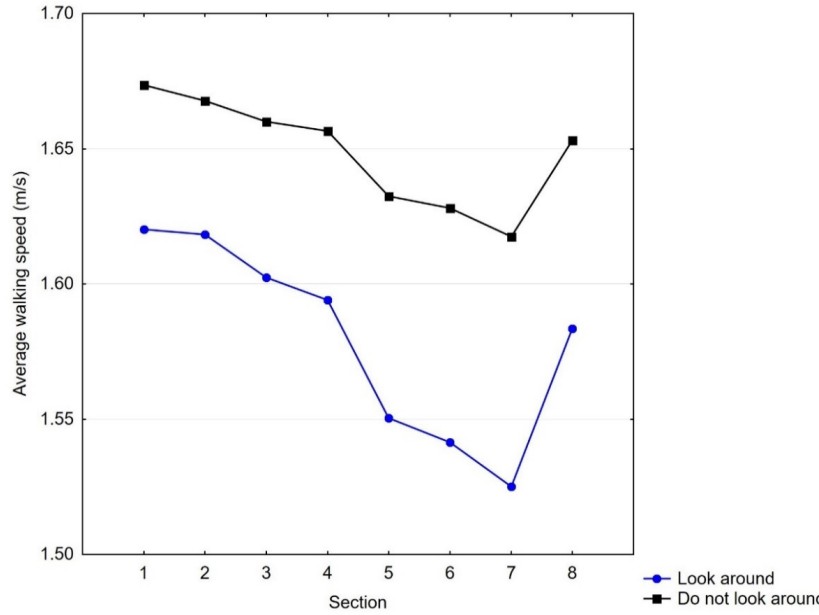

**Figure 4.** Average walking speeds (m/s) in specific sections of the route in Study 3.

## 5. General Discussion

The present study investigated the rarely explored effect of visual environmental features on pedestrian walking speed. The results showed that walking speed in specific sections of the walking route systematically differed and reflected the environmental features. While our previous studies [42,44] demonstrated the difference between walking speeds in green areas and sections without greenery, the present study was conducted on a route where all analyzed sections contained a certain amount of greenery and examined the amount of greenery within the environment combined with changes in walking speed.

The slowest walking speed was found in sections 5, 6, and 7, which were formed by a dense oak alley with mature trees surrounded by a meadow and low trees. The cognitive analysis of the visual properties of the route showed that the participants perceived a high level of space, complexity, legibility, coherence, mystery, and relaxation in these sections. These spatio-cognitive dimensions are usually combined in the environmental psychology literature with the preferred environment (e.g., [56,59,60]). These sections were also liked most. Additionally, the highest level of perceived naturalness and a low level of perceived noise and traffic were found here. Furthermore, slowing down was observed in section 10 compared to the surrounding sections 9 and 11. Importantly, sections 9, 10 and 11 were located along a road with intense car traffic. The slowing down in this section clearly demonstrated the effect of visual features compared to the effect of soundscape characteristics of the environment. In section 10, an impressive view opened up suddenly to a small park with tall old trees. Although it was located on a busy street, the perceived level of relaxation and liking was significantly higher in this section than in the surrounding sections 9 and 11.

It is worth commenting on the robustness of our findings. In addition to this study, we conducted a series of investigations on walking speed on the same route (sections 1–8) under different additional experimental conditions. In the study [66] conducted in fall 2013 and fall 2014, participants were first primed with pictures of shopping malls or trees and then walked around this route. The type of priming influenced them for all walking speeds on the route. In the study [45] conducted in spring 2017, the participants listened to traffic noise or natural sounds from headphones while walking. The type of acoustic stimulus also influenced them for all walking speeds on the route. However, under all experimental and control conditions, we observed a similar temporal pattern of walking speed: the participants gradually slowed down along sections 1, 2, 3, and 4, the slowest walking speed was in sections 5, 6, and 7, which had the highest natural character, and they subsequently walked faster in section 8.

Although we argue that the naturalness of the environment had a crucial effect on walking speed, the effect of noise should also be taken into account. In sections 5, 6, and 7, the level of perceived noise and traffic was very small because this part of the route was far from streets with traffic. However, as mentioned above, in section 10, the walking speed and the levels of relaxation and liking were significantly higher than those in surrounding sections 9 and 11, although the levels of perceived noise and traffic were equal in sections 9 and 10. However, perceived levels of noise did not refer to the actual level of noise that was objectively measured as the acoustic pressure levels in specific areas. The public available data [67] showed that the traffic noise in sections 4–8 was approximately Lday = 55–60 dBA, while the traffic noise in sections 1–3 and 9–11 was smaller, approximately Lday = 50–55 dBA. Thus, noise in areas with more greenery was subjectively perceived as smaller than it actually was, namely, in sections 5, 6, and 7. This finding is consistent with studies that found that the evaluation of sound environments can be affected by co-occurring visual settings. A natural setting makes noise less annoying (e.g., [68–70]). Moreover, it was found that greenery can mask views on streets with traffic; thus, noise may be perceived as less annoying (e.g., [71,72]). This may explain why the lowest perceived level of noise was in sections 5, 6, and 7, which were exposed to the noise of heavy traffic from a noisy, invisible motor road 300 m away with higher Lday than sections 9–11, where the perceived level of noise was significantly higher. In addition, our



previous studies [44,45] showed that acoustic stimuli were less important for pedestrians than visual stimuli because the visual features had similar effects on changes in walking speed even when the participants were listening to music, traffic noise or natural sounds.

Walking speed was the slowest in the section of the routes where the environment was liked most, and in the section where people expressed the lowest level of intention to leave the environment. Thus, the observed walking behavior may also be explained in terms of approach-avoidance behavior. These two forms of behavior are based on an affective reaction to the environment, which results in trying to establish contact with the environment (positive emotions) or intending to avoid the environment (negative emotion). Although this theory was tested in various types of service environments [50], it seems that it can also explain fluctuations in pedestrians' walking speed. Of course, when people are walking, they usually do not stop (or cannot stop) observing or joining the environment that they like, but it seems that they can unintentionally slow down. Importantly, we observed similar behavior even in the condition where the participants were asked to ignore their surroundings and just look at the road ahead.

Furthermore, it is worth commenting on why it is important to explore visual environmental factors that may slow pedestrian walking speed. A fast walk is not an undesirable behavior if it is a part of sports activities, for instance. However, the everyday walking speed of pedestrians can be associated with the overall fast pace of life as a response to certain environmental stimuli. The concept of the pace of life appeared several decades ago and has been defined as the "relative rapidity or density of experiences, meanings, perceptions and activities" ([73], p. 14). It is expected that the pace of life is a response to stimulatory overload and various urban stressors, including crowding and traffic [29]. The pace of life has been operationalized from temporal aspects of various urban behaviors and activities, e.g., mean walking speed, mean working speed, and mean clock accuracy [7]. Urban pedestrian walking speed was investigated mainly in the 1970s and the 1980s (e.g., [6,29,74]). Interestingly, Levine and Norenzayan [7] compared the pace of life, including walking speed, in large cities from 31 countries and found that the predictors for a fast pace were a higher economic level and individualistic culture, with a faster pace of city life in West European countries and Japan. This finding was also supported by a more recent study [75]. Importantly, there is evidence that a faster pace of life in large cities is associated with a greater likelihood of heart attacks [6,7]. Levine, Lynch, Miyake, and Lucia [10] explored the pace of life in metropolitan cities in the US and found a correlation between mortality rates from a coronary heart attack and a fast pace of life. Fast movement speed could be a potential risk factor for heart diseases, and it has been interpreted in parallel to Type A behavior patterns, which are associated with competitiveness, time urgency, hostility, and aggressivity. It was also shown that there is a positive correlation between an individual's pace of life and risk attitude, which can result in risky solutions to traffic situations [76].

It is worth noting that our participants walked on the route relatively quickly. Bohannon [77] compared walking speed in various age groups (20–79 years) and reported an average comfortable gait speed of approximately 1.40 m/s and maximal gait speed of approximately 2.50 m/s for males and females in their twenties. A meta-analysis of relevant studies [78] supported this finding, but the limitation of these studies was that walking speed was measured only over short distances, usually up to 10 m. In real situations, walking speed could differ according to the purpose of the walk. Our results showed that the participants walked more quickly than the reported gait speed of 1.40 m/s. Most likely, their intention was to complete the walk on the route quickly; therefore, their walking speed differed from obvious walking speed for recreation purposes when people observe and enjoy the surrounding environment. Nevertheless, walking speed in the present study varied depending on the visual environmental features of the route.

Thus, the present findings may provide a useful contribution to research on walkability. Our research has shown that natural elements on urban roads increase liking of the environment and can result in slower walking speeds. In addition, it was shown that traffic

noise, which is an environmental stressor [79], is less annoying in a green environment. Thus, natural features of a walking route may result in a decrease in tension and a feeling of time urgency. In accord with this claim, Davydenko and Peetz [80] showed that time spent walking in a natural setting was overestimated compared to actual time, in contrast to walking in a man-made setting. Creating an environment that may cope with perceived time urgency and decelerate the fast pace of life would be very useful. Thus, investigations of these phenomena have relevance.

The findings have some practical implications. Still, some architects and urban designers prefer urban design solutions with a minimal amount of greenery. For instance, they often argue that trees and greenery have not been typical in old middle age cities, that greenery gradually grows up and it may hide view on the architecture, that it occupies a space that may be used for other activities, and, moreover, that greenery needs special care, which may be expensive for municipal budgets (e.g., [81]). Although positive biological functions of urban greenery are widely known, there is less public awareness of the positive psychological effects of greenery that the current research in environmental psychology is discovering. Our findings can enhance knowledge on these positive psychological functions of urban greenery and provide architects and urban authorities more information as to why greenery can be important in new urban design solutions. Moreover, a wider awareness of the positive psychological functions of urban greenery could also help environmental organizations struggling to keep existing greenery that is considered to be removed for reasons that are not often rational.

It is worth commenting on the generalizability of our findings relative to our small samples, which consisted of young adults only. Firstly, the studies were limited to young people, which might limit the generalization of our result. There is no doubt that for comparison, it could be useful to have older people in the sample as well. On the other hand, there is no evidence for age or gender effects in responses to the natural environment (for review, see [25,82,83]). Typically, young, healthy people walk faster than older adults, but this does not mean that they perceive and react to natural environments differently than older people. Secondly, the present studies are based on an experimental approach, which limits the number of participants, because, for practical and technical reasons, it is not possible to conduct experiments with hundreds of participants. In contrast, above mentioned studies that measured urban walking speed (e.g., [7,8,29]) employed the observation method, where they measured the walking speed of a large number of pedestrians in real urban environments. However, the drawback of this approach is that it is not possible to influence the direction of their walk, to control the purpose of their walking, and other factors. However, a suggestion for future research could be to use GPS methodology that enables researchers to simply track movements of large number of people and select appropriate urban traces with diverse environments to measure walking speed. To sum up, the present study showed that people, while walking on an urban route with diverse environments, tend to slow down in the environment with a higher amount of greenery, and simultaneously, they liked this environment more in contrast to environments with a smaller amount of greenery. Although the effect of greenery on walking speed was also observed in our previous investigations [42–45], to generalize these findings requires more replications on different routes, in different cities, and even in different countries. However, many questions still remain open and cannot be successfully answered by the present study. We explained this finding with the theory of approach-avoidance behavior, proposing that people being attracted by specific environments tend to affiliate with them, explore them, and stay in them. For instance, people may be in the course of their urban walk attracted by interesting shop windows, aggressive advertisements, or by any other distinctive objects. Here, we can apply environmental psychology concepts of "hard" and "soft" fascinations [63,84,85]. According to this concept, nature represents the environment with soft fascination that is more restorative than hard fascinating environments (e.g., visit a shopping mall, sports match, watching TV, etc.). Thus, the soft fascinating function of natural elements may be to softly attract attention and elicit the need to observe natural

elements, which can influence the pace of the walk. However, this explanation needs more empirical evidence and further research. One possible way may be to use eye-tracking technology and explore what people are observing and simultaneously register their walking pace.

Secondly, given that the response to a visual environment with a specific function of greenery may be specific sensorimotor reactions, the still unanswered point is a possible synchronization of walking speed with visual structures on the route. Many studies describe rhythmical motor synchronization with auditory and visual stimuli (for a review, see [86,87]), but no one studied a possible synchronization with a "rhythm" of outdoor visual stimuli. We know that the theory of architecture often speaks about a "rhythm" of various architectural structures, such as building facades (windows, pillars, etc.) (e.g., [88]). A spatial regularity of certain architectonical features may be perceived as some form of regular rhythm. Is it possible that a walker synchronizes with the regular spatial structure of architecture in some way? Furthermore, this form of synchronization may occur in a walk in a natural environment. For instance, tree alleys usually have regular spatial arrangements; there are equal distances between individual trees. This could result in positive feelings and maybe also in slowing down because the environment is perceived as pleasant. Consistently, in the present study, we observed the slowest walking speed in sections 5–7 formed by the alley. In contrast, an irregular spatial arrangement may result in dissatisfaction and an unconscious intention to leave the environment. Clearly, this point needs future research.

Our research also has further limitations. Firstly, in any experiment conducted in real outdoor conditions, it is difficult to control all variables. We could not control immediate changes in atmospheric conditions or changes in traffic. Although the route was not frequently used by other walkers, our participants sometimes met other pedestrians. Therefore, we divided the experiment into one-hour blocks and we randomly assigned the participants to the experimental conditions. Additionally, the video recordings were carefully checked, and data were removed when a participant did not respect the rules of the experiment (for instance, he/she stopped) or was disturbed by anything on the route.

Another limitation might be that the study was conducted only in one vegetation period in early spring. We do not know whether people would react in the same way in different vegetation periods, for instance, in late fall when the trees have no foliage or in wintertime with snow. It is also worth noting that Study 1 and Study 2 were conducted in 2013, while Study 3 was conducted in 2016. However, other than small changes in vegetation, the route appearances did not differ substantially.

## 6. Conclusions

This study explored the effects of the environmental features of an urban route on walking speed. It was found that walking speed was slower in the environment with a higher level of naturalness. This environment was also more liked, and noise, which is an urban environmental stressor, was perceived as less annoying in this environment than in an environment with a lower number of natural elements. The results are in accordance with environmental preference research that has documented various benefits of walking in the natural environment. The addition of green elements on urban walking routes can increase positive feelings from a walk, provide a greater walk experience, and reduce stress and tension. The findings also have some practical implications for architects, urban designers, and the actions of environmental activists and organizations.

**Supplementary Materials:** The following are available online at https://www.mdpi.com/article/10.3390/land10050459/s1, Table S1: dataset.

**Author Contributions:** M.F. and L.R. conceived and designed the experiments; M.F. and L.R. performed the experiments; M.F. and L.R. analyzed the data; M.F. wrote the paper. All authors have read and agreed to the published version of the manuscript.

**Funding:** The study was supported by the Student Specific Research Grants 1/2021 and 3/2021 from the Faculty of Informatics and Management at the University of Hradec Králové.

**Institutional Review Board Statement:** The study was conducted according to the guidelines of the Declaration of Helsinki, and approved by the Committee for Research Ethics at the University of Hradec Králové, No. 6/2020, 23 November 2020.

**Informed Consent Statement:** Informed consent was obtained from all subjects involved in the study.

**Data Availability Statement:** The datasets supporting this article have been uploaded as part of the Supplementary Materials.

**Acknowledgments:** We thank Tereza Barková, Vít Brouček, Ondřej Chyba, and Jan Petružálek for their help in organizing and conducting the experiments.

**Conflicts of Interest:** The authors declare no conflict of interest.

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
