# Peer review of "Environmental Features Influence Walking Speed: The Effect of Urban Greenery"

_land, doi:10.3390/land10050459_

Round 1
Reviewer 1 Report
- Indicate the type of road pavement and whether it is always the same.
- Explain the choice of the group of riders based on age.
- It would be interesting to know if there are significant differences in terms of walking speed in the each zones between males and females also for the other parameters evaluated.
Author Response
Dear reviewer, thank you for your comments.
Point1: Indicate the type of road pavement and whether it is always the same.
Response1: This was indicated, an asphalt pavement, line 206.
Point2: Explain the choice of the group of riders based on age.
Response2: Psychological experiments are usually conducted on university students. It is supposed that some basic functions of perceptual and cognitive systems are common of for people various generations, thus, using university students do not bias results. To explain this selection and point out differences between general population, we included a paragraph to the General discussion, lines 726-737. Typically, young, healthy people walk faster than older adults, but it does not mean that they perceive and react to natural environments differently than older people. There is no evidence for age or gender effects in responses to the natural environment [for review, see 25, 82, 83].
Point3: It would be interesting to know if there are significant differences in terms of walking speed in the each zones between males and females also for the other parameters evaluated.
Response3: In preliminary data analysis we also conducted an ANOVA with gender as the predictor, but gender had not any significant effect.
Reviewer 2 Report
Abstract and Introduction
I think that the key objective of the paper is clear: "The objective of the present study was to investigate the effect of visual environmental features on pedestrian walking speed." But in the introduction, you can try to answer some points of interest. 1. An effective introduction answers three sets of questions: a. Who cares? What is the topic or research question, and why is it interesting and important in theory and practice? b. What do we know, what don't we know, and so what? What major, unaddressed puzzle, controversy, or paradox does this study address, and why does it need to be addressed? c. What will we learn? How does your study fundamentally change, challenge, or advance scholars' understanding?
You mention: “While our previous studies investigated the contrast between the effects of visual features of a built urban environment without greenery, a built urban environment with greenery, and an urban natural environment, the current study aims to explore the effect of visual features of urban environments where all sections of the environment contain various levels of greenery.” This review of his previous studies is of little significance in presenting his analysis, he should be more specific about what this work contributes to the scientific literature.
In the abstract, I would include the methodology used and not the details of the sample.
Theoretical framework
Although it is comprehensive, a deeper literature review is needed. When studying the visual characteristics of the environment and their influence on the speed of the walker, previous research cited is outdated, there are few references and it is recommended to include research done in the last years, It includes more than 80 references but only 7 belong to the years 2019 and 2020.
There is one detail of the study that is not clear to me, it focuses on environments related to the environment but there could be other elements that also slow down walkers and no mention is made of them: landscape architecture, the destination of the path, visual elements that attract walkers such as advertisements, etc. Explain why they have not been included.
Methodology
When using ANOVAs in your study, it is striking that you do not mention starting hypotheses, can you explain why and why you do not use other methods of analysis. It is also important that you mention the statistical software you have used for your study.
Results and discussion: It is recommended to indicate the program that is used for statistical analyses.
Conclusion: You must improve this section, it is the weakest of your work. As your findings are interesting ones, you can suggest a conceptual model based on them. This conceptual model can be tested in future research. However, you can at least discuss and develop some propositions based on the relationships of your conceptual model. In your Conclusion, you do not discuss theoretical and managerial implications, though you discuss limitations and future research.
Finally, it could be interesting to highlight the practical significance for organizational members of this study and make the reference to policy prescriptions that derive from this analysis, as well as the implications for future research. I will encourage the authors to expand the agenda for future research.
Author Response
Dear reviewer, thank you for evaluation of our work and constructive comments.
Please see the attachment.

Reviewer 3 Report
The research direction of the paper is based on a nice idea, which is to investigate how visual environmental features can influence walking speed. The article is well structured, with a clear research orientation and very interesting results. The bibliographic references included in the article are numerous and satisfactorily cover its research field.
The main weaknesses of the article are related to the methodology used and its scientific "depth". I think the methodology is relatively simple and has no particular scientific interest. This is a critical point, because, I believe that the methodological contribution of the article is rather restricted.
In addition, a comparison of the final results found in the article with the final results of other similar surveys would serve both their better understanding and using in the future.
Author Response
Dear reviewer, thank you for the positive evaluation of our work. Here are responses to your comments:
Point1: The main weaknesses of the article are related to the methodology used and its scientific "depth". I think the methodology is relatively simple and has no particular scientific interest. This is a critical point, because, I believe that the methodological contribution of the article is rather restricted.
Response1: We employed the method that was used in the previous studies that explored urban walking speed. Those authors used human observers, who registered the walk time with stopwatchs. In contrast, we used more sophisticated method, the video recording of the walk and subsequent analyzing of walking speed. Yes, there are studies that explored pedestrian flow on sidewalks or cross-roads, which used more sophisticated methods and mathematical models, but we think that our methodology is for the goals of these experiments appropriate and sufficient. But to respect your constructive criticism, in the general discussion, we are discussing directions for future research and recommend to combine the measurement of walking movements with registering of eye movements via eye-tracking methodology (line 760-762), or use GPS methodology in analyses of walking speed of urban dwellers (line 739-742).
Point 2: In addition, a comparison of the final results found in the article with the final results of other similar surveys would serve both their better understanding and using in the future.
Response2: The problem is that no one conducted a similar research. There is only in limited number of studies exploring pace of life including walking speed, mainly from the 1970s and the 1980s.
Reviewer 4 Report
This is a well designed and well executed study. The authors provide sufficient framing of the study through a well developed literature review. The methodology is clearly explained. The multiple studies within the paper are clearly analyzed. Appropriate lessons are drawn that are supported by the evidence. The authors close with useful recommendations for policy or practice. This is a worthy paper for this journal.
Author Response
Dear reviewer, thank you for the positive evaluation of our work.
Reviewer 5 Report
Walking speed depends on many factors, e.g., the destination or the purpose of the movement as well as its obligatory nature. The conditions and environment in which this trip takes place are also important. The authors of the manuscript focused on investigating the influence of urban greenery on the speed of walking. I find it as a very interesting research subject.
The structure of the manuscript is correct. The authors cite many literature sources, which I consider very valuable. I highly appreciate the comparative analysis to the results of other studies carried out in the discussion section.
The authors applied their own method, dividing the research into three parts:
- Study 1 - analysis of walking speed on eleven selected sections of the walking route, leading through areas consisting of city parks and built-up areas with lots of greenery,
- Study 2 - assessment of the environmental characteristics of the walking route by the participants of the study.
- Study 3 - analysis of the impact of the purpose of trip on walking speed.
I have no objections to the research method, but the sample size, in my opinion, is too small to be representative. Such studies can only be considered as a pilot study.
Due to the fact that the research sample was limited to young people (aged 19 to 28 years), I suggest to emphasized it in the title of the article.
In addition, taking into account the length of sections of a specific type, I believe that they are too short to formulate general conclusions and dependencies.
It is also worth conducting research on a larger number of walking routes.
Author Response
Dear reviewer, thank you for the positive evaluation of our work. Here are responses to your comments:
Point1: I have no objections to the research method, but the sample size, in my opinion, is too small to be representative. Such studies can only be considered as a pilot study.
Response1: We added discussion about this limitation given by the sample size to General discussion, see lines 733-742. From practical and technical reasons, psychological experiments cannot have a large sample of hundred participants. We are discussing different research approaches that could work with large samples as a recommendation for future research.
Point2: Due to the fact that the research sample was limited to young people (aged 19 to 28 years), I suggest to emphasized it in the title of the article.
Response2: We emphasized this information in Abstract and added extensive discussion to General discussion, lines 726-732. We were not able to find any good form of the title informing that our participants were young people. On the other hand, to stress the young participants in the title might lead to the belief that the described mechanisms work only in young people. But we can't say that because we didn't have older participants in the sample.
Point3: In addition, taking into account the length of sections of a specific type, I believe that they are too short to formulate general conclusions and dependencies.
Response3: In previous investigations, even shorter length of measured sections were used. For instance, Levine & Norenzayan (1999) in “The pace of life in 31 countries” measured the walking speed in distance of 60 feet only. Further studies used similar short distances. In contrast, in our study we measured longer distances, such as 60 – 100 meters. It would be difficult to measure walking speed in longer sections, because environmental features may change, or turns may appear on the route, etc.
Point4: It is also worth conducting research on a larger number of walking routes.
Response4: We added that as a recommendation for a future research, lines 746-749
Round 2
Reviewer 2 Report
Many thanks to the authors for the new manuscript, which reflects the suggestions made for the improvement of the work presented.
Congratulations.
Kind regards.